# Job Attributes and Mental Health: A Comparative Study of Sex Work and Hairstyling

**Bill McCarthy** [1,*], **Mikael Jansson** [2] **and Cecilia Benoit** [2]

1    School of Criminal Justice, Rutgers University Newark, 123 Washington Street, Newark, NJ 07102-3094, USA
2    Canadian Institute for Substance Use Research, University of Victoria, 2300 McKenzie Ave., Victoria,
     BC V8N 5M8, Canada; mjansson@uvic.ca (M.J.); cbenoit@uvic.ca (C.B.)
*    Correspondence: wm307@scj.rutgers.edu

**Abstract:** A growing literature advocates for using a labor perspective to study sex work. According to this approach, sex work involves many of the costs, benefits, and possibilities for exploitation that are common to many jobs. We add to the field with an examination of job attributes and mental health. Our analysis is comparative and uses data from a panel study of people in sex work and hairstyling. We examined job attributes that may differ across these occupations, such as stigma and customer hostility, as well as those that may be more comparable, such as job insecurity, income, and self-employment. Our analysis used mixed-effects regression and included an array of time-varying and time-invariant variables. Our results showed negative associations between mental health and job insecurity and stigma, for both hairstyling and sex work. We also found two occupation-specific relationships: for sex work, limited discretion to make decisions while at work was negatively related to mental health, whereas for hairstyling, mental health was positively associated with self-employment. Our results highlight the usefulness of an inter-occupational labor perspective for understanding the mental health consequences of being in sex work compared to hairstyling.

**Keywords:** sex work; mental health; job attributes; job insecurity; stigma; service work; hairstyling





## 1. Introduction

A growing literature examines prostitution as sex work (see the review in Benoit et al. 2019). This perspective argues that as an economic activity, selling sex is grounded in commodification and the exchange of various types of capital (e.g., erotic and economic). The exchange of sexual services reflects, in part, the set of economic opportunities available to those who sell, as well as the market forces that contribute to demands for specific activities and workers (Constable 2009; O'Connell Davidson 2014; West and Austin 2002; Zelizer 2005). According to this labor approach, selling sex can be usefully studied as a type of service work comparable to other personal service jobs, particularly those that involve emotional labor, body work, and related activities (Sullivan 2010).

Researchers have used a labor perspective to study sex work in a wide variety of economic contexts and geographic locations. Much of this research uses ethnography and qualitative interviews to obtain a deep understanding of the occupational experiences of relatively small samples of people who sell sex (e.g., Rosen and Venkatesh 2008; Sanders 2005). Like other service jobs, sex work presents several avenues for exploitation and mistreatment from employers, customers, law enforcement, and the general public. Studies suggest that this exploitation and mistreatment may have negative consequences for mental health (Krumrei-Mancuso 2017; Puri et al. 2017; Treloar et al. 2021; Vanwesenbeeck 2005), but like most other areas in sex work research, these studies typically draw on small samples and focus exclusively on people working in the sex industry.

One exception is a recent intra-comparative study of sex work that involved face-to-face interviews with a relatively large sample ($n$ = 218) surveyed in six metropolitan areas of Canada (Benoit et al. 2020b). Most respondents had worked in several personal

service jobs across their careers, and one-third said they were currently employed outside of sex work. Their other work was largely part-time, gig-economy work, or temporary jobs in personal services such as retail, restaurant, and hospitality work. They choose among several competing alternatives when navigating the employment opportunities available to them, moving into and out of a variety of personal service jobs. The majority said that sex work gave them relatively more personal satisfaction, greater control over their working conditions, and higher earnings. The main challenges of sex work, compared to their other jobs, were low prestige and stigma.

We add to this research with an *inter-comparative* study of two types of work that scholars have variously called routine interaction, frontline, or personal service (Cohen et al. 2013): sex work and hairstyling or barbering (hairstyling hereafter). As we explain below, these two types of work differ in many ways but share a number of features that warrant a comparative analysis. Our data are taken from a panel survey study of a relatively large sample of adults aged 19 or over, working for more than one year in one of two places, Victoria, British Columbia (Canada), and Sacramento, California (USA). The hairstyling sample included people with earnings from cutting or styling hair in a shop or salon, including those who were self-employed and rented a chair in a salon, owned a salon, or cut hair at their home. The sex work sample was limited to people who exchanged a sexual service for money, had one-to-one, direct physical contact with customers, and were paid directly by them. We chose these criteria to focus on the experience of sex workers who provided sexual services to clients in person on at least a part-time basis, in a parallel way that stylists provide in-person services to their clients. These interactions occur in a number of contexts ranging from street soliciting to off-street work in escorting and erotic massage businesses.

Our study focuses on connections between work attributes and self-reported mental health. There is an extensive literature on the connections between job conditions and mental health (e.g., Benach et al. 2014). We draw on this literature and focus on characteristics assumed relevant across a wide array of occupations—including employment insecurity and job decision making—as well as features that may be especially pertinent for the two occupations we consider, such as self-employment, customer hostility, and stigma.

### 1.1. Sex Work and Hairstyling: Similarities

Sex work and hairstyling share many features with other frontline service jobs. In this paper, we focus on job insecurity, income, prestige, emotional labor, and self-employment. In general, frontline service jobs are often described as bad jobs or precarious work, in part because they are typically characterized by high levels of job insecurity (Kalleberg 2011; Sherman 2007). Indeed, Kalleberg (2009, p. 17) argues that job insecurity, and precarious work more broadly, is "the dominant feature of the social relations between employers and workers in the contemporary world."

Although there are exceptions, frontline service work generally offers low earnings and little prestige. Data from the early 2000s showed that more than half of the top 30 jobs most frequently found among the United States (U.S.) working poor involved frontline service work (Wicks-Lim 2012). Data from the U.S. Bureau of Labor Statistics (2018) indicated that the median income for styling was $24,730 USD. In Canada, hairstyling and barbering are among the 10 occupations with the lowest income, with median yearly employment incomes of approximately $21,305 CAD for full-time, full-year workers (Statistics Canada 2018).

There are much less data on the income of sex workers, and the existing data are not easily comparable to the income records for jobs in the formal economy (Parent et al. 2013). A study of street-based female sex work in Vancouver, Canada, reported median weekly earnings from sex work of $300 CAD, or approximately $15,600 CAD a year (Deering et al. 2011); however, the study reported considerable variation in income, with a quarter of respondents reporting that they earned less than $100 CAD a week (~$5200 CAD a year) and a quarter reporting that they earned $560 CAD a week or more (~$29,200 CAD).

A heterogeneous sample of sex workers aged 19 and over in five Canadian census metropolitan areas found the median total income (earnings and government transfers) for people in sex work was $39,500 CAD compared with $34,204 CAD for the general Canadian population aged 15 and over (Benoit et al. 2020b) but was much lower compared with the employment income (wages, salaries, and commissions plus net self-employment income) of full-time, full-year workers aged 15 and over ($53,431 CAD; Statistics Canada 2018). Hairstyling (Nakao and Treas 1994) has low occupational prestige and is at the low end of socio-economic status (SES) scales. Sex work is typically excluded from occupational prestige and SES status research (see Hauser and Warren 1997).

Another similarity between sex work and hairstyling is that they both demand what has been variously referred to as emotion work, aesthetic labor, or soft skills (Dwyer 2013; Hochschild 1983; Witz et al. 2003). Jobs of this nature require workers to manage their own and their customers' feelings to create positive interactions and increase the likelihood of receiving a generous tip and repeat customers/clients (Cohen 2010; Hill and Bradley 2010; Sanders 2005). Effortful emotional regulation involves both surface and deep acting (Grandey 2000); the former highlights workers' emotional expressions that are primarily a response to encounters with customers, whereas the later include workers' felt, as well as expressed, emotions that can occur prior to, as well as during, customer interactions. The centrality of emotion work in hairstyling and sex work suggests that both economic activities are psychologically demanding and that they often place workers in a position in which customers may have considerable control over the workers' decisions and actions.

Hairstyling and sex work also share parallels in terms of self-employment. Although some stylists work as employees (e.g., in chain salons), many stylists are self-employed, typically working as independent contractors who rent space (chairs) in salons (Black 2004; Cohen 2010). Sex work may also involve employees, as is the case in some escort agencies and massage parlors, but most work involves what some researchers have identified as independent contractors or entrepreneurs (Smith and Christou 2009). In research conducted in Great Britain, Pitcher (2015) found that many of the workers she spoke with described their work in ways that corresponded with accounts from self-employed workers in other jobs: people described having the freedom to set their own hours, services, fees, and working conditions (e.g., location). Sex work may also involve combinations of various types of income generation. People may independently advertise their services directly to customers through various media outlets (Benoit et al. 2017) while working concurrently as contractors for escort agencies or as employees in massage parlors. This flexibility may serve people well should they lose employment with a particular agency or find that some type of work is no longer viable (e.g., due to changes in legal codes).

*1.2. Sex Work and Hairstyling: Differences*

These two frontline service jobs also differ in important ways. Here we draw attention to the backgrounds of people who selfselect into the two types of labor and four work attributes: job insecurity, legal restrictions, stigma, and customer hostility. There is an extensive literature on the backgrounds of people involved in sex work (see Benoit et al. 2019 for a review), but most of this work draws exclusively on samples of people who previously had worked, or were working, in the sex industry (e.g., Rosen and Venkatesh 2008; Sanders 2005). Only a few studies compared the backgrounds of people working in the sex industry with those of other groups, and many of these focused on groups of drug users (e.g., Maher 1997; Mosack et al. 2010) or victims of childhood abuse (e.g., Wilson and Widom 2010). Prior research that analyzed the comparative data we use showed that sex work is associated with a variety of limiting and negative experiences (McCarthy et al. 2014): compared to workers in other frontline service occupations, including hairstyling, people in sex work were less likely to have completed high school, had worked in fewer occupations, and had more extensive use of some illegal drugs (e.g., cocaine). They were also more likely to have experienced childhood poverty and abuse.

Sex work also involves more legal risks than hairstyling (McCarthy et al. 2012). Although hairstyling is illegal in some contexts (e.g., cutting hair without a license), there are few restrictions compared to prohibitions of a wide array of sexual services. Prohibitions also vary by place, as is evident in the laws from the two places where the data we use in this study were collected. In the U.S., sex work is governed mostly by state law. The California Penal Code, at the time of the study and continuing to today, criminalizes accepting money or other compensation for sex. It penalizes soliciting as well as engaging in prostitution as misdemeanors that, after the first conviction, result in mandatory jail time. In Canada, sex work is mostly controlled though the federal criminal code. At the time the data for this study were collected, it was legal in Canada to sell and buy sexual services, although it was almost impossible to do so without breaking laws prohibiting solicitation and bawdy houses (Morton et al. 2012).

Stigma is a major determinant of health (Goffman 1963; Hatzenbuehler et al. 2014). Sex work is highly stigmatized, in part because of moral, religious, and ideological arguments against it (Benoit et al. 2019, 2020b), even where it is decriminalized and regulated (Abel and Fitzgerald 2010). Many occupations carry a stigma, especially those that involve the body (Black 2004) or dirty work (Ashforth and Kreiner 1999; Hughes 1962). Hairstyling may, therefore, carry some occupational stigma; however, this stigma may be less intense and consequential than that associated with sex work (Benoit et al. 2018).

Additionally, sex work may have a greater level of customer hostility and aggression. A growing body of research finds that customer incivility occurs in a number of service settings (Wilson and Holmvall 2013). Ethnographic studies of hairstyling, for example, report that some customers, especially those who are from higher-status backgrounds, expect subservience and mistreat those who cut their hair (Black 2004; Hill and Bradley 2010; Robertson and O'Reilly 2020). However, comparative research suggests that customer hostility is infrequent in hairstyling. In a national study of US workers (Alterman et al. 2013), personal care work was 11th out of 22 occupations in terms of workplace hostility ("During the past 12 months, were you threatened, bullied, or harassed by anyone on the job?"). In contrast, customer hostility, aggression, and violence are commonly reported by workers in the sex sector (Benoit et al. 2018; Deering et al. 2014; Sawicki et al. 2019; Strega et al. 2020).

*1.3. Work and Mental Health*

There is wide and diverse literature that connects many of the job attributes we examine with mental health. This literature consistently shows that mental health is negatively associated with job demands and positively associated with job control. Meta-analyses and systematic reviews of the literature show strong associations between mental health and income inequality (Patel et al. 2018), job insecurity (Benach et al. 2014; Kim and von dem Knesebeck 2015; Lee et al. 2018; Sverke et al. 2002), and stigma (Link and Phelan 2001; Mak et al. 2007). Individual studies show strong relationships between mental health and other job attributes such as psychological demands and limited opportunities to make decisions (Mausner-Dorsch and Eaton 2000; Paterniti et al. 2002; Ten Have et al. 2015). Scholars have speculated that customer hostility will have negative consequences for mental health (Sliter et al. 2010), whereas self-employment could have positive effects (Hessels et al. 2017; Nikolova 2019; however, see Rietveld et al. 2015), but there is relatively little research on either relationship.

Collectively, the foregoing findings raise several questions about the relationship between job attributes and mental health. First, are the well-documented associations between mental health and an array of job attributes, such as job insecurity and earnings, similar in analyses of hairstyling and sex work? Second, do self-employment and other job attributes that are especially common in these occupations contribute to mental health in meaningful ways? Third, are associations between mental health and various personal service job attributes similar or different for sex work and hairstyling? The analysis we present below examines these questions.

## 2. Materials and Methods

### 2.1. Sample and Data Collection

We assessed our research questions with four waves of data gathered in a unique panel study that ran from 2003 to 2008 in two urban areas: the census metropolitan area of Victoria, British Columbia (Canada), and one of three counties that belong to the greater metropolitan area of Sacramento, California (USA). The research protocol was approved by the Institutional Review Board and the Human Research Ethics Committee of the authors' universities (University of California Davis and University of Victoria, respectively)). The study focused on adult (aged 19 and over) frontline service workers whose primary paid work involved working in one of three occupations, two of which we examine here: sex work and hairstyling (the third, food and beverage serving, did not have enough self-employed workers to be included in this analysis). Workers had to have one-to-one, direct physical contact with customers to be considered for the study. Given this parameter, sex work in our study included activities such as escorting, erotic massage, and on-street soliciting but excluded work in which there is no direct physical contact with clients (e.g., phone sex or media productions). We asked participants to name the job title they used to describe their work. For those in sex work, the most common responses were escort, prostitute, sex worker, and sex trade worker. We provide only a brief summary of the study here because an extended discussion is available elsewhere (Jansson et al. 2013).

One of the most prominent challenges to occupation-based studies involves the absence of high-quality population lists. This study addressed this problem by using a combination of random and purposive sampling techniques. These included the following: contacting, by phone and mail, a random sample of barbers and stylists selected from a California state list of licensees; contacting, in person, by phone or mail, managers of hairstyling, food-and-beverage-serving, and sex industry businesses to post a study advertisement in their businesses; posting study advertisements in local newspapers; and using respondent-driven sampling (Heckathorn 2002) in which respondents recommended the study to other workers and received a small incentive if their contacts participated. We used these approaches to draw samples that included a diverse array of workers and that could provide a basis for generalizing to the phenomena studied (Luker 2010).

The study obtained a first-wave sample of 212 people in sex work and 181 who cut and styled hair. A greater percentage of workers in hairstyling were recruited in Sacramento than in Victoria (64% versus 35%), whereas the reverse occurred for people in sex work (55% in Victoria versus 45% in Sacramento).

We interviewed respondents four times, with a period of at least four months between interviews. At each of the study's four waves, the research team gathered information with an interviewer-completed questionnaire, an interviewer-guided but respondent-completed survey, and a structured interview. Retention rates for the second through fourth waves were higher for people who worked in styling (90%, 89%, 87%) compared to those for people in sex work (72%, 61%, 55%). We examined attrition with a multinomial logit model of attrition at each wave; first-wave involvement in sex work predicted attrition at the second and third waves, while race, and alcohol and marijuana use predicted attrition at wave two (for all participants). We included these variables in our analyses.

### 2.2. Measures

#### 2.2.1. Dependent Variable: Mental Health

This study examined overall mental health with data from each of the four waves of the study (see Table A1 for item details). We used answers to two questions ($r$; W1 = 0.66, W2 = 0.65, W3 = 0.73, and W4 = 0.66; $p < 0.001$) for our overall mental health scale. These questions asked respondents to use rank-ordered answer categories to rate their mental health (1 = Poor . . . 5 = Excellent) and how often they had been feeling unwell mentally (1 = Always/chronically . . . 5 = Never) in the four months preceding each interview. General questions like these are common in health research, and a review of 57 studies found that they are associated with measures of mental health that use a larger battery of items

(Ahmad et al. 2014). We standardized our mental health scales and the other scales we used in our analysis by dividing them by the number of items.

2.2.2. Independent Variables
Job Attributes

We measured seven job attributes. Two of these, self-employment and income, were assessed at each wave of the study. At each wave, respondents reported whether they were currently self-employed (1 = Self-employed 0 = else) and what their income was for the last month from wages and tips (logged in our multivariable analysis). Time-varying variables allowed for an assessment of the effects of differences between respondents, as well as those for within-person variation over time.

Our remaining job attribute measures were collected only once, at the second wave. Thus, we were able to examine relationships between these job attributes and changes in mental health over time, but we could not examine connections between mental health and changes in these job attributes. Our first time-invariant variables focused on customer hostility and stigma. We measured the former with a single question that asked respondents about their agreement with the following statement: "I am subject to hostility or abuse from clients or customers" (1 = Strongly disagree . . . 4 = Strongly agree). Our stigma measure was a nine-item scale ($\alpha$ = 0.71; 95% CI (0.66, 0.77)) that used revised versions of items commonly used to measure enacted and felt stigma (Scambler 2009). Unlike other approaches (e.g., Wahl 1999), our questions did not connect discrimination directly to a specific attribute (e.g., mental illness, race) or to sex work. Instead, we used questions that allowed us to estimate the association between stigma and mental health for a variety of occupations. Two questions asked respondents how often they had been refused rental housing they could afford or were denied a bank loan (1 = Never . . . 5 = Very often; respondents skipped these questions if they indicated they had not applied for rental housing or a loan). Two items asked respondents how often doctors or nurses said things about their occupations, and one item asked how frequently people looked down on them (1 = Never . . . 5 = Very often). Five items asked respondents about agreement with the statements about doctors and nurses treating them respectfully and their family and others treating them respectfully (1 = Strongly agree . . . 5 = Strongly disagree).

Our final three job attribute variables concerned job insecurity, decision latitude, and psychological demands. We measured these with items from the Job Content Questionnaire (JCQ; http://www.jcqcenter.org/). The JCQ measures have high reliability across occupations, demographic attributes, and countries (Karasek et al. 1998). The JCQ has been widely used to study work and mental health in the formal economy (e.g., de Jonge et al. 2000; Wang et al. 2008), and some researchers have demonstrated its usefulness for studying occupations in the informal employment sector (e.g., de Araújo and Karasek 2008).

Our JCQ measure of job insecurity was based on four questions ($\alpha$ = 0.71; 95% CI (0.66, 0.76)). These asked respondents about the following: (1) their perceptions of the likelihood that they could lose their job at some point in the next two years (1 = Not at all likely, 2 = Not too likely, 3 = Somewhat likely, 4 = Very likely); (2) the steadiness of their current job (1 = Regular and steady, 2 = Seasonal, 3 = Frequent layoffs, 4 = Both seasonal and frequent layoffs); (3) whether their job was secure (1 = Strongly agree . . . 4 = Strongly disagree); and (4) prior exposure to layoffs (1 = Never, 2 = Faced the possibility once, 3 = Faced the possibility more than once, 4 = Constantly or actually laid off).

We measured decision latitude with the JCQ eight-item scale ($\alpha$ = 0.82; CI (0.78, 0.85)) that has two components, skill discretion and decision authority. Skill discretion focused on the possibilities for creativity and the extent to which a job maximizes workers' skills. Decision authority highlighted autonomy and opportunities to make decisions. We included five questions that measure skill discretion (e.g., "My job requires me to be creative") and three for decision authority (e.g., "On my job, I have very little freedom to decide how I do my work"). Each question used Likert-style responses (1 = Strongly disagree . . . 4 = Strongly agree).

The third JCQ measure, psychological demands, was a five-item scale ($\alpha$ = 0.48; 95% CI (0.38, 0.57); dropping items does not increase $\alpha$) that assessed mental workload and conflicting demands and constraints (Karasek et al. 1998) suggested that the lower internal consistency for this scale in some studies may reflect the greater subjectivity of its items, relative to those in the other scales). The questions in this scale asked about contradictory demands and having to do work that is repetitive, fast, hard, or excessive (1 = Strongly disagree . . . 4 = Strongly agree).

### 2.2.3. Control Variables

Our analysis included a diverse set of time-varying and time-invariant predictor variables linked to mental health (Elo 2009; Umberson et al. 2010). We included six time-varying variables that refer to the four months prior to each interview. The first two concerned health-enhancing behaviors and contact with health care professionals. The former was a count of behaviors (1 = Yes, 0 = No), from a list of six, that respondents reported doing to improve their health (e.g., exercising more, changing the diet or eating habits, quitting or reducing smoking). The latter was a count (0 = No, 1 = Yes) of types of medical services, from a list of four, used at each wave (e.g., family physician, hospital emergency care). We included a control for living with a romantic partner (0 = No, 1 = Yes) and one for the number of children who lived with the respondent (includes biological, adopted, and children of partners, relatives, or others). The remaining two time-varying variables used an ordinal scale to measure substance use (0 = Never, 1 = Less than once a month, 2 = Twice a month, 4 = Once a week, 8 = Twice a week, 30 = Daily or more). The first variable (substance use 1), a two-item scale, focused on alcohol and marijuana use, whereas the second (substance use 2), a five-item scale, concerned the use of club drugs (such as ecstasy), non-prescribed prescription drugs (such as OxyContin), crystal methamphetamine, cocaine, and heroin.

We measured our time-invariant control variables at the first wave. They included five demographic variables. Age was measured with an open-ended question. We used closed-ended questions with a category Other for the remaining four variables: gender identification (0 = Male, 1 = Female; seven respondents chose Other and were excluded from this analysis), sexual orientation (0 = Heterosexual, 1 = Other (Homosexual, Bisexual, Other)), race (0 = White, 1 = Non-White), and nativity (0 = Native born, 1 = Born outside of Canada/United States). The analyses also included dummy variables related to socio-economic standing. We used answers to two questions (i.e., highest-completed grade, post-secondary education/training) to construct four dummy measures of education (i.e., less than high school, high school graduate (comparison category), some college, and completed college). Our unemployment variable measured unemployment (1 = Yes, 0 = No) at waves two through four.

We included two variables that measured childhood experiences linked to adult health (Elo 2009). The first was a dichotomous measure of childhood sexual abuse. The second was a scale measure of childhood economic hardship (i.e., a three-item scale; $\alpha$ = 0.80; 95% CI (0.75, 0.84)), based on items that asked about parental/guardian difficulties paying for necessities, school expenses, and recreational activities (1 = Rarely/Never . . . 5 = Almost always/Always). The analyses also comprised dummy variables for health insurance (0 = No, 1 = Yes) and country of residence (0 = Canada 1 = U.S.). Prior comparative analyses of health inequities in Canada and the U.S. underscore the importance of both variables (e.g., Prus 2011; Siddiqi et al. 2009). All of the Canadian respondents reported that they belonged to a government health care insurance program, while 58% of our US respondents said they had health insurance. The majority of the latter had coverage through a private plan they purchased or obtained through their or their romantic partners' employment; the remainder received government-provided care. Estimates for the U.S. suggest that less than half of US workers had an employer-subsided health care plan in the mid-2000s (Haley-Lock 2011, p. 828).

Variance inflation factor scores for our independent variables were all less than three, indicating that collinearity is not an issue. Between 10% and 15% of respondents' information for some of our independent variables was missing, and so we used multiple imputation (with 30 data sets) to impute missing data for missing values (von Hippel 2007). The majority of missing cases occurred among respondents who were doing sex work at the first wave of the study. We did not impute values for any cases that were missing information about our dependent variables at *any wave*; this resulted in an analytical sample of 273.

### 2.3. Analysis

Our analysis began with an examination of descriptive statistics and mean and percentage differences between workers in sex work and hairstyling for our key variables. We then turned to mixed-effects linear regression (Allison 2009). Mixed-effects linear regression combined the fixed-effects vector decomposition approach (Bell and Jones 2015) that uses individuals as their own controls (to help address issues associated with time-stable unobserved heterogeneity) and the random-effects approach for dealing with time-invariant variables such as demographic attributes and several of our job attribute measures. Mixed-effects models also decomposed the effects of time-varying variables into two components: One was estimated with a deviation score (centered time-varying variables around the person-specific means) and captured within-person variation, whereas the second was based on the person-specific mean and captured between-person differences. Our between-person component was comparable to cross-sectional results; it examined, for example, whether drug use was negatively associated with mental health, on average. The within-person component measured changes across the four waves of data, the same as fixed-effects models, and investigated, for instance, whether, on average, a person's mental health changed if his/her substance use changed.

## 3. Results

### 3.1. Descriptive Statistics

Demographic data for our analytical sample showed similar gender distributions across sex work and hairstyling: 87% and 82% of respondents, respectively, identified as women (see Table 1). In contrast, the two groups differed significantly on most other demographic attributes. A greater proportion of people in sex work identified as non-heterosexual (42% and 12% respectively) and as a member of a racial or ethnic minority (37% and 21%); in contrast, a greater proportion of people in hairstyling said they were immigrants (16% and 7%). People in sex work were also younger on average than those in hairstyling (average age 37 and 44); a greater percentage did not attend high school (47% compared to 15%), and a smaller percentage did not complete their secondary education (34% compared to 59%).

**Table 1.** Descriptive Statistics, Test of Difference, and Effect Size, Control Variables +.

| Variable | Sex Work | | Styling | | | Hedges' |
|---|---|---|---|---|---|---|
| | Mean | SD | Mean | SD | t | G |
| *Predictors* | | | | | | |
| Health care use | | | | | | |
| Wave 1 | 0.30 | 0.23 | 0.24 | 0.22 | 1.93 | 0.27 |
| Wave 2 | 0.31 | 0.18 | 0.21 | 0.20 | 3.41 *** | 0.52 |
| Wave 3 | 0.33 | 0.19 | 0.21 | 0.22 | 3.74 ** | 0.58 |
| Wave 4 | 0.25 | 0.17 | 0.19 | 0.20 | 1.94 | 0.32 |
| Health enhancing behaviors | | | | | | |
| Wave 1 | 0.72 | 0.45 | 0.76 | 0.43 | 0.68 | 0.09 |
| Wave 2 | 0.70 | 0.46 | 0.67 | 0.47 | 0.58 | 0.06 |
| Wave 3 | 0.78 | 0.44 | 0.70 | 0.47 | 1.38 | 0.17 |
| Wave 4 | 0.62 | 0.49 | 0.70 | 0.46 | 1.40 | 0.17 |
| Living with spouse/partner | | | | | | |
| Wave 1 | 0.22 | 0.42 | 0.58 | 0.50 | 6.35 *** | 0.77 |
| Wave 2 | 0.29 | 0.46 | 0.58 | 0.50 | 4.93 *** | 0.60 |
| Wave 3 | 0.32 | 0.47 | 0.58 | 0.50 | 4.53 *** | 0.53 |
| Wave 4 | 0.34 | 0.48 | 0.58 | 0.50 | 3.93 *** | 0.50 |
| Number of children living with respondent | | | | | | |
| Wave 1 | 0.42 | 0.76 | 0.68 | 0.99 | 2.41 * | 0.29 |
| Wave 2 | 0.44 | 0.80 | 0.64 | 1.02 | 1.81 | 0.21 |
| Wave 3 | 0.48 | 0.89 | 0.68 | 0.99 | 1.36 | 0.21 |
| Wave 4 | 0.49 | 0.95 | 0.65 | 1.09 | 1.35 | 0.15 |
| Substance use 1 | | | | | | |
| Wave 1 | 8.37 | 8.89 | 5.34 | 7.26 | 2.93 * | 0.38 |
| Wave 2 | 5.90 | 7.48 | 4.93 | 7.02 | 1.06 | 0.13 |
| Wave 3 | 6.54 | 8.31 | 4.75 | 6.82 | 1.87 | 0.24 |
| Wave 4 | 5.52 | 8.06 | 4.78 | 7.16 | 0.77 | 0.10 |
| Substance use 2 | | | | | | |
| Wave 1 | 4.16 | 6.09 | 0.11 | 0.52 | 6.59 *** | 1.04 |
| Wave 2 | 2.87 | 4.29 | 0.18 | 0.89 | 6.33 *** | 0.96 |
| Wave 3 | 2.79 | 4.59 | 0.19 | 0.97 | 5.59 *** | 0.84 |
| Wave 4 | 2.20 | 3.79 | 0.10 | 0.58 | 5.73 *** | 0.85 |
| Age | 37.35 | 8.93 | 44.32 | 13.05 | 5.24 *** | 1.19 |
| Childhood poverty | 1.80 | 1.03 | 1.52 | 0.78 | 2.46 ** | 0.31 |
| Childhood abuse | 3.44 | 6.36 | 0.55 | 1.36 | 4.80 *** | 0.67 |

| Variable | Sex Work | Styling | | Hedges' |
|---|---|---|---|---|
| | % | % | Chi Sq | G |
| Gender (female) | 86.84 | 82.05 | 1.13 | 0.14 |
| Non-heterosexual | 42.47 | 12.33 | 31.50 *** | 0.74 |
| Racial minority | 37.17 | 21.29 | 8.18 ** | 0.33 |
| Immigrant | 6.90 | 15.92 | 5.12 * | 0.28 |
| Less than high school | 46.55 | 14.64 | 33.53 *** | 0.76 |
| High school graduate | 33.63 | 58.60 | 16.68 *** | 0.52 |
| Some college | 12.07 | 17.20 | 1.38 | 0.14 |
| Completed college | 7.75 | 9.55 | 0.27 | 0.07 |
| Unemployed | 12.93 | 1.91 | 13.16 *** | 0.45 |
| Health insurance | 83.62 | 85.99 | 0.59 | 0.06 |

+ based on original data; *** $p < 0.001$, ** $p < 0.01$, * $p < 0.05$ (two-tailed).

Table 2 shows descriptive statistics for mental health and job attributes. The means at each wave suggested the average respondent had between good (=3) and very good (=4) overall mental but that people employed in hairstyling reported, at all four waves, significantly better mental health compared to people in sex work. Self-employment was common across both sectors; it was more prevalent in sex work than in hairstyling at the first wave (84% versus 69%) but less frequent in the former in other waves (e.g., 44% in sex work versus 61% in hairstyling at wave four). The decline in self-employment

over time likely reflected a number of patterns, including the greater fluidity between self-employment and employed labor in sex work. Changes in policing practices to focus on street selling and improved opportunities for employment at massage parlors, escort agencies, and managed settings could have also played a role. There was also greater sample attrition of independent workers in sex work relative to those who worked for others.

Earnings were similar across occupations, except for the fourth wave, when earnings for hairstyling were significantly greater than those for sex work. Across the four waves, the average monthly income for respondents from Canada ranged from $1566CAD (wave 4) to $3138CAD (wave 1) in sex work and from $2106CAD (wave 3) to $2426CAD (wave 2) in hairstyling. For respondents from the U.S., the average monthly income ranged from $2375USD (wave 4) to $3387USD (wave 1) in sex work, and from $3032USD (wave 2) to $3376USD (wave 1) in hairstyling. Sex work had significantly higher levels of customer hostility, stigma, and job insecurity on average but also significantly greater decision latitude than hairstyling.

**Table 2.** Descriptive Statistics, Test of Difference, and Effect Size, Mental Health and Job Attributes +.

| Variable | Sex Work | | Styling | | | Hedges' |
|---|---|---|---|---|---|---|
| | Mean | SD | Mean | SD | t | G |
| Overall mental health | | | | | | |
| Wave 1 | 3.17 | 1.02 | 3.73 | 0.87 | 7.71 *** | 0.60 |
| Wave 2 | 3.25 | 1.05 | 3.63 | 0.89 | 7.73 *** | 0.40 |
| Wave 3 | 2.99 | 1.07 | 3.62 | 0.97 | 4.80 *** | 0.62 |
| Wave 4 | 3.00 | 1.08 | 3.54 | 0.85 | 4.20 *** | 0.56 |
| Job attributes | | | | | | |
| Income (logged, in thousands) | | | | | | |
| Wave 1 | 3231.80 | 2791.53 | 2922.21 | 2181.70 | 0.74 | 0.13 |
| Wave 2 | 2542.14 | 2329.89 | 2798.05 | 2145.11 | 1.84 | 0.12 |
| Wave 3 | 2529.44 | 2555.94 | 2812.95 | 2031.74 | 1.53 | 0.13 |
| Wave 4 | 1919.37 | 1631.73 | 2855.38 | 2181.38 | 3.40 *** | 0.46 |
| Customer hostility | 2.93 | 0.88 | 2.16 | 0.80 | 7.36 *** | 0.92 |
| Stigma | 2.38 | 0.62 | 1.81 | 0.29 | 8.32 *** | 1.24 |
| Insecurity | 2.10 | 0.74 | 1.54 | 0.58 | 6.22 *** | 0.87 |
| Decision latitude | 1.94 | 0.48 | 1.57 | 0.40 | 6.61 *** | 0.85 |
| Psychological demands | 2.78 | 0.44 | 2.85 | 0.40 | 1.41 | 0.17 |

Percentage Tests of Differences

| Variable | Sex Work | | Styling | | | Hedges' |
|---|---|---|---|---|---|---|
| | % | SD | % | SD | Chi Sq | G |
| Self-employed | | | | | | |
| Wave 1 | 0.84 | 0.37 | 0.69 | 0.46 | 7.84 ** | 0.35 |
| Wave 2 | 0.65 | 0.48 | 0.71 | 0.45 | 0.28 | 0.13 |
| Wave 3 | 0.52 | 0.50 | 0.66 | 0.45 | 4.71 * | 0.30 |
| Wave 4 | 0.44 | 0.50 | 0.61 | 0.49 | 7.92 ** | 0.34 |

+ based on original data; *** $p < 0.001$, ** $p < 0.01$, * $p < 0.05$ (two-tailed).

### 3.2. Multivariable Analyses

Our multivariable analyses highlighted several patterns (see Table 3). In both sex work and hairstyling, the unstandardized coefficients for job insecurity were significant and negatively related to mental health ($b = -0.25$, se = 0.10 and $b = -0.35$, se = 0.10, respectively). A *Z*-test of the difference between the coefficients (Clogg et al. 1995) was not significant, suggesting that job insecurity has comparable consequences for the mental health of workers in each occupation. A similar pattern occurred for stigma: it was significantly and negatively associated with mental health for both sex work and hairstyling ($b = -0.26$, se = 0.13 and b = $-0.38$, se = 0.16, respectively), and the coefficients for the two occupations were not significantly different in their magnitude.

**Table 3.** Mixed Effects Multivariable Regression, Overall Mental Health and Job Attributes.

| Variables | Sex Work | | Styling | |
|---|---|---|---|---|
| | *b* | se | *b* | se |
| Self-employed (within) | −0.23 | (0.24) | 0.34 * | (0.14) |
| Self-employed (between) | 0.06 | (0.11) | 0.32 ** | (0.12) |
| Income (within) | 0.06 | (0.03) | 0.02 | (0.04) |
| Income (between) | 0.03 | (0.05) | 0.01 | (0.06) |
| Customer hostility | −0.01 | (0.09) | 0.10 | (0.06) |
| Stigma | −0.26 * | (0.13) | −0.38 * | (0.16) |
| Insecurity | −0.25 * | (0.10) | −0.35 *** | (0.10) |
| Little decision latitude | −0.41 * | (0.19) | 0.06 | (0.14) |
| Psychological demands | −0.30 | (0.17) | −0.13 | (0.14) |
| Health enhancing behaviors (within) | −0.02 | (0.10) | −0.08 | (0.07) |
| Health enhancing behaviors (between) | 0.17 | (0.28) | 0.16 | (0.18) |
| Health care use (within) | −0.37 | (0.25) | −0.21 | (0.18) |
| Health care use (between) | −1.23 ** | (0.41) | −1.05 ** | (0.37) |
| Living with spouse/partner (within) | −0.06 | (0.16) | 0.22 | (0.21) |
| Living with spouse/partner (between) | −0.06 | (0.18) | 0.14 | (0.11) |
| Number of children living with (within) | −0.01 | (0.08) | −0.02 | (0.06) |
| Number of children living with (between) | −0.06 | (0.11) | −0.02 | (0.05) |
| Substance use 1 (within) | −0.02 ** | (0.01) | 0.00 | (0.01) |
| Substance use 1 (between) | −0.00 | (0.01) | −0.01 | (0.01) |
| Substance use 2 (within) | −0.03 | (0.01) | −0.04 | (0.05) |
| Substance use 2 (between) | 0.01 | (0.02) | 0.02 | (0.10) |
| Age | −0.01 | (0.01) | 0.00 | (0.00) |
| Gender (1 = female) | −0.10 | (0.20) | −0.16 | (0.13) |
| Sexual minority | 0.08 | (0.16) | 0.06 | (0.16) |
| Racial minority | 0.07 | (0.15) | 0.23 | (0.13) |
| Immigrant | 0.02 | (0.31) | −0.14 | (0.15) |
| Non high school graduate | 0.12 | (0.15) | −0.08 | (0.15) |
| Some college | 0.62 * | (0.25) | 0.07 | (0.14) |
| College graduate | 0.45 | (0.28) | 0.23 | (0.17) |
| Ever unemployed (W1 to W4) | −0.15 | (0.21) | −0.50 | (0.37) |
| Childhood economic insecurity | −0.14 * | (0.06) | −0.08 | (0.07) |
| Childhood abuse | −0.00 | (0.01) | −0.04 | (0.04) |
| Access to health care | 0.49 * | (0.23) | 0.25 | (0.16) |
| Country (USA = 1) | 0.68 ** | (0.21) | −0.11 | (0.12) |
| Wave 2 dummy | −0.08 | (0.10) | −0.13 | (0.07) |
| Wave 3 dummy | 0.20 | (0.11) | −0.12 | (0.08) |
| Wave 4 dummy | 0.11 | (0.12) | −0.19 * | (0.08) |
| Constant | 5.84 *** | (0.91) | 4.89 *** | (0.77) |
| Number of observations/groups | 347/93 | | 593/155 | |

+ multiple imputation data (30 times); *b* = unstandardized regression coefficient, se = standard error; *** $p < 0.001$, ** $p < 0.01$, * $p < 0.05$ (two-tailed).

Another commonly measured job attribute, limited opportunities to make decisions, was also significantly and negatively associated with mental health but only for people engaged in sex work ($b = -0.41$, se = 0.19). Our results indicated that although sex work has higher levels of customer hostility and more intense psychological demands than hairstyling, neither variable was predictive of mental health, the net of the other variables in our model. Our results also highlighted the returns of self-employment for mental health but only for hairstyling. Self-employment in this sector was significantly and positively associated with mental health ($b = 0.32$, se = 0.12), and mental health improved for stylists who became self-employed during the course of the study ($b = 0.34$, se = 0.14).

We illustrated the strength of the associations between cognitive job security and health with regression-predicted values in Figures 1 and 2, holding all other variables constant at their means. The slope of each line indicates the strength of the relationship, with steeper slopes indicating stronger associations. In Figure 1, the graph for job insecurity

and stigma in sex work shows that people with low job insecurity and limited exposure to stigma had predicted mental health scores that were about 30% higher than the scores for people with the highest job insecurity and stigma exposure. Differences in the decision-making latitude while working were also consequential: the predicted mental health score was about 60% higher for people who had a high level of decision making compared to those who reported low levels of it.

The graphs for job insecurity and stigma for hairstyling follow similar patterns to those for sex work (see Figure 2). The predicted mental health scores were approximately 38% higher for people with low job insecurity compared to those whose job insecurity was high and about 40% higher for people who reported low levels of stigma compared to those with reported high levels of it. Results for self-employment indicated predicted mental health scores that were 1.36 points higher for self-employed stylists compared to those who were employees in a salon or shop (results not graphed). The results for changes in the employment status indicated that someone who became self-employed during the study improved his/her predicted mental health score by 1.20 points.

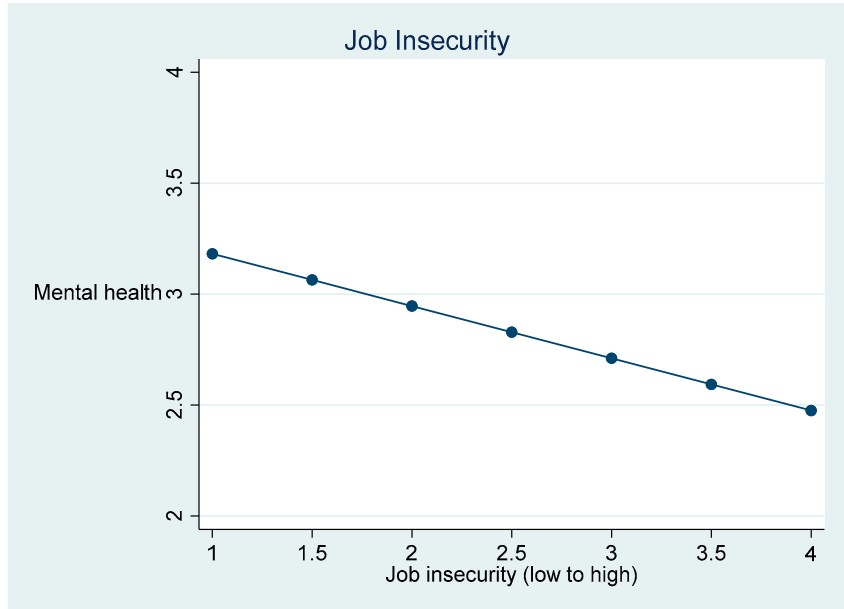

**Figure 1.** *Cont*.

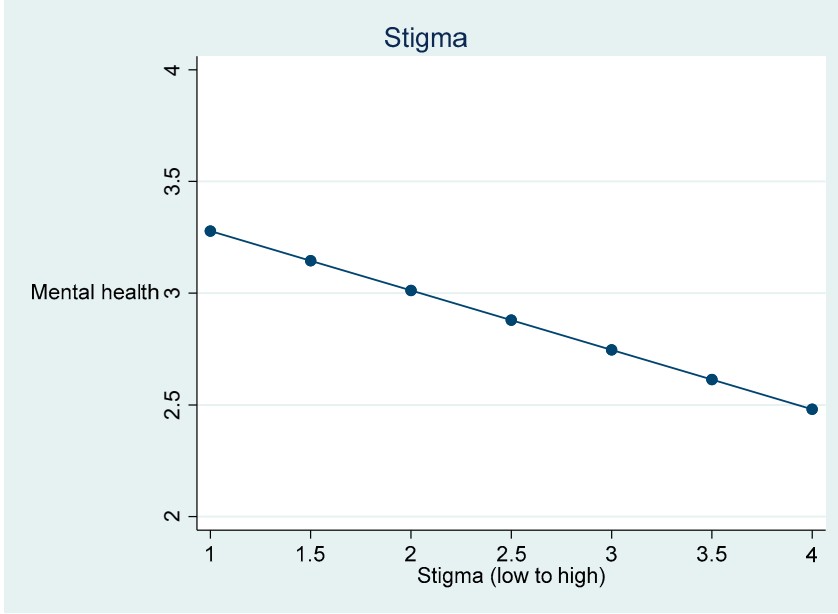

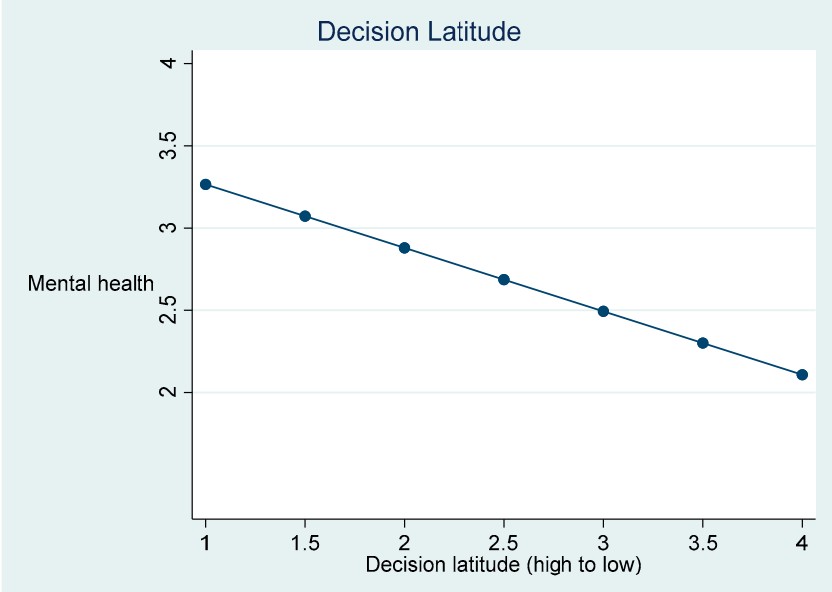

**Figure 1.** Predicted values, mental health, and job attributes in sex work.

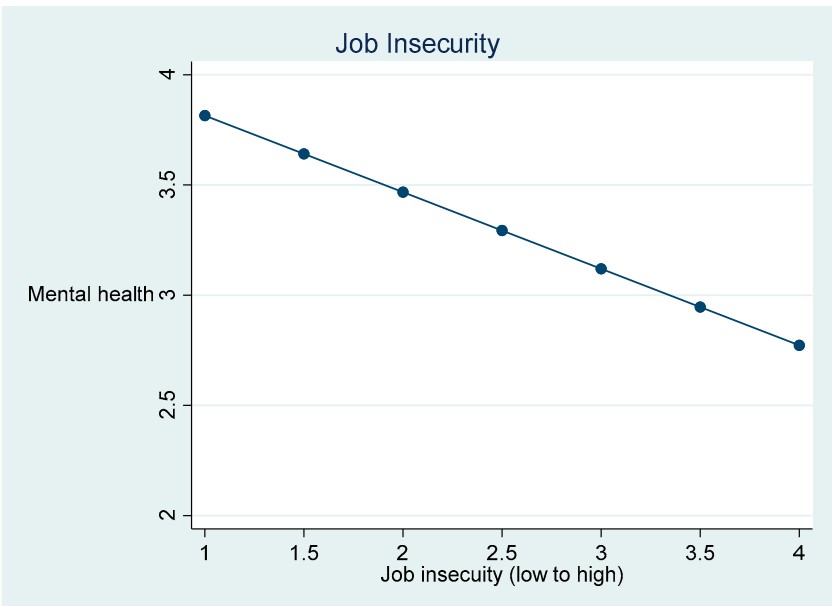

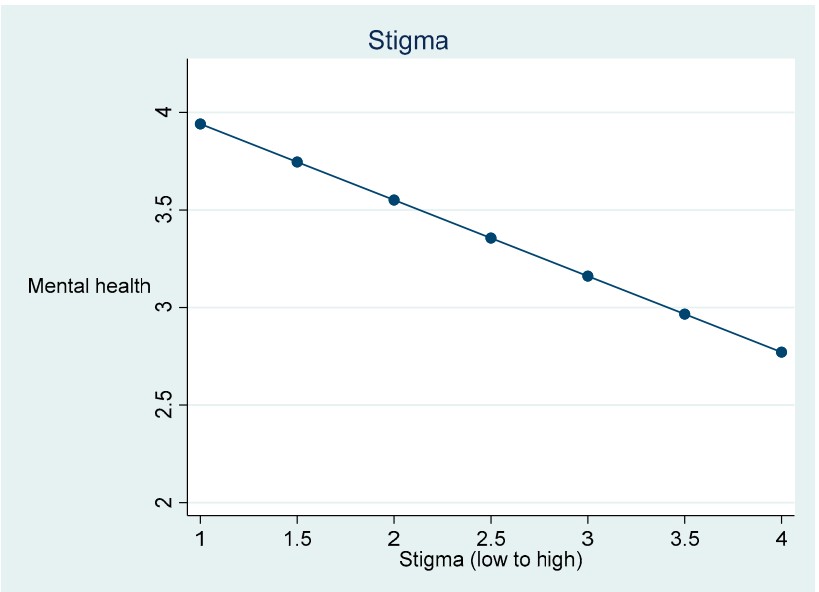

**Figure 2.** Predicted values, mental health, and job attributes in hairstyling.

## 4. Discussion

Jobs are a focal point for much of people's lives, and it is no surprise that they have important consequences for mental health. Studies of a broad array of occupations have shown that positive job features enhance our mental health and well-being, while negative attributes diminish them (e.g., Lee et al. 2018; Patel et al. 2018). Yet, this research has largely ignored sex work. There is a growing literature showing the value of examining sex work as an economic activity that involves the performance of various forms of labor for economic returns (Benoit et al. 2017, 2020b; Rosen and Venkatesh 2008; Sanders 2005). Meanwhile, several studies have suggested links between sex work and mental health ( Krumrei-Mancuso 2017; Puri et al. 2017; Treloar et al. 2021; Vanwesenbeeck 2005) but have focused on stigma and mistreatment by others and have typically ignored work attributes that are not uniquely associated with sex work (Benoit et al. 2016).

Our study builds on our previous research that compared sex work to other jobs sex workers were doing concurrently (Benoit et al. 2020b) and that compared it to related

occupations (Benoit et al. 2015a, 2015b). This study added to those studies by examining a variety of work attributes common to two frontline service jobs: hairstyling and sex work.

Our analyses provided several findings. We found that job insecurity is associated with poorer mental health in both sex work and hairstyling. This finding resonates with an extensive literature on the link between job insecurity and mental health, especially among jobs that are often described as low-income frontline work, jobs that account for an ever-increasing share of work in neo-liberal capitalist markets (Kalleberg and Vallas 2018; Olsthoorn 2014). The robust association between job insecurity and worse mental health underscores the need to address it as a serious public health and social concern.

We also found that another commonly used measure of job attributes, skill discretion, was important for understanding connections between working conditions and poor mental health (e.g., Paterniti et al. 2002; Ten Have et al. 2015). Our results showed that this pattern is pronounced for sex work but not for hairstyling. The associations that we documented between mental health and job insecurity and limited decision making for sex work provide compelling evidence of the usefulness of using insights from labor research and the sociology of work and occupations to understand sex work (Benoit et al. 2019).

Our research also showed that the negative relationship between stigma and mental health commonly described in studies of people in sex work (Benoit et al. 2018) is not unique to that occupation; instead, we found that the mental health of stylists is worse for those who report high levels of stigma (Benoit et al. 2019). This finding suggests that although sex work carries a more intense stigma than hairstyling, the negative consequences of being stigmatized are similar for these two occupations and, indeed, may be similar across a wide array of occupations that involve dirty work (Abel 2011; Ashforth and Kreiner 1999; Benoit et al. 2020a; Hatzenbuehler et al. 2014).

Our last finding concerns self-employment. There is relatively little research on the relationship between self-employment and mental health. The results of prior research are equivocal; some studies found that the self-employed report better mental health (Nikolova 2019) compared to wage or contract workers, whereas others found little difference between the two groups once adjustments were made for backgrounds and selection (e.g., Rietveld et al. 2015). However, prior research has typically used aggregate data from a broad array of occupations. Our study demonstrates that the mental health benefits of self-employment may be pronounced for certain occupations, such as hairstyling, and negligible for others, such as sex work.

*Future Research*

Our paper has a number of strong features—four waves of panel data, a broad set of job measures, and the estimation of both between- and within-person effects—but is not without limitations that we hope future research can address. The study combined random and non-random sampling techniques, and so our samples of people working in styling and sex work may not be representative. The job measures we used were not collected at all four waves of the study, and some, like customer hostility, were based on a single question. There are now alternatives for measuring customer incivility that include more items (Wilson and Holmvall 2013). Mental health may be influenced by a number of other job attributes that we did not consider, such as relationships with managers and coworkers. Recent research has found that managers of sex work businesses operate in ways that are similar to other personal service industries (Casey et al. 2017; Horning and Marcus 2017), and managers' behaviors in these settings may have effects similar to those documented in research on other occupations (e.g., McDonough et al. 2008).

Our study assumed that work influences mental health, but it is possible that some other unmeasured confounder variables influence the jobs people choose and their views about them. In mixed-effects analyses, people act as their own controls, thereby reducing the effects of omitted variable bias. However, these analyses cannot eliminate the possibility that pre-existing, unmeasured conditions may influence views about one's job. Our study does not include measures of the larger political and economic context in which

employment and job instability are embedded. In most neo-liberal capitalist economies, including the U.S. and Canada where our samples were collected, service jobs continue to grow in number and the structure and character of this work continue to evolve with the changing economic climate.

Another limitation is that our data were collected more than a decade ago. The occupational structure in the U.S. and Canada may have changed the relationship between job attributes and mental health, some that may have affected only people in sex work and others that may also have had an impact on people employed in hairstyling. Increased media, political attention, and laws targeting sex trafficking have likely changed how sex work operates in both countries (Global Network of Sex Work Projects (NSWP)). In Canada, new prostitution laws enacted in 2014 ban the purchase of sexual services and the receipt of material benefits from prostitution and procuring services, make it illegal for newspaper and magazine publishers, website administrators, and web-hosting services to publish advertisements for any sexual services, and prohibit communicating for the sale and purchase of sexual services in a public place next to a school ground, playground, or daycare center (Department of Justice Canada 2014). These changes are likely to further impact the nature of sex work in Canada and their consequences for the mental health of sex workers.

Finally, hairstyling and sex work have likely both been influenced by the stagnation of wages that continued into the 2010s for service work and by the economic downturn and disappearance of work that occurred as a result of the 2020 COVID-19 pandemic. Subsequent research will need to consider these and other changes in assessing the relationships between work attributes and mental health.

Despite these limitations, this is, to the best of our knowledge, the only inter-comparative quantitative study of the mental health impacts of a relatively large and diverse sample of adults engaged in sex work compared to people in hairstyling. While there is a rich literature on the connections between job conditions and mental health (e.g., Benach et al. 2014), our study fills in an important gap in the connections between work attributes and self-reported mental health for two gig-economy jobs that are absent from this literature.

## 5. Conclusions

Researchers have used a variety of approaches to studying sex work, treating it as inherently exploitative, a crime, and a morally unacceptable act, as well as seeing it as a type of labor characterized by many of the same demands, exploitations, costs, and benefits of other paid work. Our results highlight the usefulness of the latter, inter-occupational labor perspective for understanding the mental health consequences of being in sex work compared to hairstyling. This approach is also relevant for reforming laws that regulate sex work, designing social policies that benefit workers, and developing strategies to combat sex work stigma.

**Author Contributions:** Conceptualization; methodology; investigation; supervision; project administration; resources; funding acquisition: C.B., M.J., and B.M. equally; formal analysis; data curation: B.M. and M.J. equally; writing—original draft preparation: B.M.; writing—review & editing: C.B. and M.J. equally. All authors have read and agreed to the published version of the manuscript.

**Funding:** This research was funded by the Canadian Institutes of Health Research, the Canadian Studies Grant Program, the University of California Davis, and the University of Victoria.

**Institutional Review Board Statement:** The study was conducted according to the guidelines of the Declaration of Helsinki, and approved by the Institutional Review Board of the University of California Davis (project no. 2004-12138) and the Human Research Ethics Committee of The University of Victoria (project no. 257-02).

**Informed Consent Statement:** Informed consent was obtained from all subjects involved in the study.

**Data Availability Statement:** Please contact the authors about data availability.

**Conflicts of Interest:** The authors declare no conflict of interest.

## Appendix A

**Table A1.** Variable Coding.

| **Health (measured waves 1–4)** | |
| --- | --- |
| Overall mental health | In the last four months how would you rate your mental health? 1 = Poor 2 = Fair 3 = Good 4 = Very good 5 = Excellent; How often have you been feeling unwell mentally? 1 = Always/chronically 2 = Very often 3 = Sometimes 4 = Not often 5 = Never |
| **Job Attributes (measured wave 2)** | |
| Self-employed | Are you currently self-employed? 0 = No 1 = Yes |
| Income | Logged monthly income from wages/tips from primary job |
| Insecurity | (1) Sometimes people permanently lose jobs they want to keep. How likely is it that during the next couple of years you will lose your present job with your employer? 1 = Not at all likely 2 = Not too likely 3 = Somewhat likely 4 = Very likely; (2) How steady is your work? 1 = Regular and steady 2 = Seasonal 3 = Frequent layoffs 4 = Both seasonal and frequent layoffs; (3) During the past year, how often were you in a situation where you faced a job loss or layoff? 1 = Never 2 = Faced the possibility once 3 = Faced the possibility more than once 4 = Constantly or actually laid off; (4) My job security is good 1 = Strongly agree 2 = Agree 3 = Disagree 4 = Strongly disagree |
| Little decision latitude | (1) My job requires that I learn new things; (2) My job requires me to be creative; 3) My job allows me to make a lot of decisions on my own; (3) My job requires a high level of skill; (4) On my job, I have very little freedom to decide how I do my work (reverse coded); (4) I get to do a variety of different things on my job; (5) I have a lot of say about what happens on my job; (6) I have an opportunity to develop my own special abilities 1 = Strongly agree 2 = Agree 3 = Disagree 4 = Strongly disagree |
| Psychological demands | (1) My job requires working very fast; (2) My job requires working very hard; (3) I am free from conflicting demands that others make (reverse coded); (4) I am not asked to do an excessive amount of work (reverse coded); (5) My job involves a lot of repetitive work 1 = Strongly disagree 2 = Disagree 3 = Agree 4 = Strongly agree |
| Customer hostility | I am subject to hostility or abuse from clients or customers. 1 = Strongly disagree 2 = Disagree 3 = Neither agree of disagree 4 = Agree 5 = Strongly agree |
| Stigma | (1) I have applied for, but have been refused an apartment when I could afford it; (2) I have applied for, but I have been refused a bank loan or other credit; (3) How often do doctors say anything about the work you do? (4) How often do nurses say anything about the work you do? 1 = Never 2 = Seldom 3 = Sometimes 4 = Often 5 = Very often (5) Doctors usually treat me with respect (reverse coded); (6) Nurses usually treat me with respect (reverse coded); (7) People think I am an intelligent person (reverse coded); (8) My family accepts me as I am (reverse coded); (9) People shy away from me when they get to know me; 1 = Strongly disagree 2 = Disagree 3 = Agree 4 = Strongly agree |
| **Control Variables** | |
| **Time-varying (measured waves 1–4)** | |
| Health enhancing behaviors | 1 point (to a maximum of 6) for each of the following: exercising more, quitting or reducing smoking, drinking less alcohol, changing diet or eating habits, taking vitamins, or other, unlisted behaviors |
| Health care use | 1 point (to a maximum of 4) for each of the following: physician, physician specialist, psychiatrist, or hospital emergency care |
| Living with spouse/partner | 0 = No 1 = Yes |
| Children | Number of children living with respondent |
| Substance use 1 | How often in the last four months did you consume (1) alcohol (2) marijuana? 0 = Never, 1 = Less than once a month, 2 = Twice a month, 4 = Once a week, 8 = Twice a week, 30 = Daily or more |
| Substance use 2 | How often in the last four months did you consume (1) cocaine; (2) club drugs (e.g., ecstasy); (3) non-prescribed prescription drugs (e.g., OxyContin); (4) crystal methamphetamine; (5) heroin? 0 = Never, 1 = Less than once a month, 2 = Twice a month, 4 = Once a week, 8 = Twice a week, 30 = Daily or more |

**Table A1.** *Cont.*

| Time-invariant (measured wave 1) | |
|---|---|
| Gender | 0 = Male 1 = Female |
| Sexual minority | 0 = No 1 = Yes |
| Racial minority | 0 = No 1 = Yes |
| Nativity | 0 = Native 1 = Immigrant |
| Non high school graduate | 0 = No 1 = Yes |
| High school graduate | 0 = No 1 = Yes |
| Some college | 0 = No 1 = Yes |
| College graduate | 0 = No 1 = Yes |
| Ever unemployed | 0 = No 1 = Yes (at any time during study) |
| Access to health care | 0 = No 1 = Yes |
| Country | 0 = Canada 1 = U.S. |
| Childhood economic insecurity | While you were growing up, were your parents/guardians able to pay for: (1) Basic necessities (like food, clothing or rent); (2) Things you needed for school (like school supplies, going on local field trips, etc.); (3) Recreational activities (like playing soccer or other sports, movies, eating out, vacation, or music lessons)? 1 = Rarely/never 2 = Some of the time 3 = Half of the time 4 = Most of the time 5 = Almost always/always |
| Childhood abuse | Count of the number of years (from birth to 18) that the respondent was the victim of sexual abuse |

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
