# Peer review of "Job Attributes and Mental Health: A Comparative Study of Sex Work and Hairstyling"

_socsci, doi:10.3390/socsci10020035_

Round 1

Reviewer 1 Report

The paper is well written and researched as well as being novel in using a labour framework to compare sex work, hairstyling and mental health.

The data reported in the study is from 2003-2008 and at more than 10 years old, there is a question about currency that the authors need to address in the paper. Are the findings still valid after more than 10 years? It is more than 10 years since collection and sex work is a very fluid and constantly changing, and practices can become outdated over that period of time. Sex work in Canada has radically transformed since 2008, is this also the case for the US with the shift to viewing sex work as a form of sex trafficking under the TIP Acts? The authors need to address and comment on this, and raise it as a major limitation of the study.

More needs to be said about the type/types of sex work participants engaged in. It appears that the authors are making claims/assumptions based on full-service sex work, but this is not made clear. Sex work encompasses more than full-service sex work and the authors need to be clear about the types of sex work research participants did. A major issue here is a claim made that income data for sex workers was among the lower half of income distribution, however, this is not the case for all sex workers – some sex workers have the highest incomes earned and are among the highest half of income distribution, so more clarity is needed. The claim about sex work having low occupational prestige is backed by a very old study from 1980 and is premised on the view of sex work as criminal/a deviant activity. This stands in contradiction to the authors claim that sex work is viewed in the paper as and through a labour lens. Suggest revising this and finding more suitable and current studies and that the authors may wish to undertake a more nuanced and critical analysis.

The authors need to be clear about the types of sex work as this is necessary in enabling an understanding of the comparisons made, but also in understanding the importance of laws and legal contexts and so in addition to the above, the authors need to consider the different legislative contexts. Some consideration is given to this, but this is cursory and not enough to fully understand the impacts this may have on workers’ experiences. A major question here is that the US seems to be statistically significant (as indicated in Table 3) – could the different laws be a factor? Sex work is illegal in the US, but this was not the case in Canada until 2014 when buying sex was criminalised. The authors need to include a section about the sex work laws in the two different countries, and this needs to include the relevant laws during the periods of data collection. This needs to be taken into account in the analysis as this is likely to have a significant impact on workers’ mental health and job insecurity. Here the authors also need to give consideration to the point that not all forms of sex work are full-service (but this can still include direct physical contact) and thus other forms of sex work may be legal in the two research sites where full-service sex work may be criminalised.

Suggest re-labelling the ‘socially more and less acceptable drug use’ categories to use terms that are less judgmental (e.g., alcohol and marijuana consumption; illicit and non-prescription drug consumption). The manuscript needs a copy edit, there are a few typos and errors with citations and in the reference list etc. And, detail needs to be provided on the HREC/IRB/IEC review body that approved the project.

Author Response

We thank the reviewers for their helpful suggestions and we have revised our paper accordingly. We provide our responses below, in italics.

Reviewer 1:

The paper is well written and researched as well as being novel in using a labour framework to compare sex work, hairstyling and mental health.

1) The data reported in the study is from 2003-2008 and at more than 10 years old, there is a question about currency that the authors need to address in the paper. Are the findings still valid after more than 10 years? It is more than 10 years since collection and sex work is a very fluid and constantly changing, and practices can become outdated over that period of time. Sex work in Canada has radically transformed since 2008, is this also the case for the US with the shift to viewing sex work as a form of sex trafficking under the TIP Acts? The authors need to address and comment on this, and raise it as a major limitation of the study.

We have added the following text on page 29 (and references) to address these points:

  Another limitation is that our data were collected more than a decade ago. The occupational structure in the U. S. and Canada may have changed the relationship between job attributes and mental health, some that may have affected only people working in sex work, and others that may also have had an impact on people employed in styling. Increased media, political attention, and laws targeting sex-trafficking have likely changed how sex work operates in both countries (Global Network of Sex Work Projects, 2011). In Canada, new prostitution laws enacted in 2014,  ban the purchase of sexual services, receipt of material benefits from prostitution and procuring services, and make it illegal for newspaper and magazine publishers, website administrators, and web-hosting services to publish advertisements for any sexual services; they also prohibit and prohibits communicating for the sale and purchase of sexual services in a public place next to a school ground, playground, or daycare center (Department of Justice Canada, 2014). These changes are likely to further impact the nature of sex work in Canada and its consequences for the mental health of sex workers.

Finally, styling and sex work have likely both been influenced by the stagnation of wages that continued in to the 2010s for service work and by the economic downturn and disappearance of work that occurred as a result of the 2020 COVID-19 pandemic.  Subsequent research will need to consider these and other changes in assessing the relationships between work attributes and mental health. 

Despite these limitations, this is the only inter-comparative quantitative study of the mental health impacts of a relatively large and diverse sample of adults engaged in sex work compared to people in styling. While there is a rich literature on the connections between job conditions and mental health (e.g., Benach et al., 2014), our study fills in an important gap in the connections between work attributes and self-reported mental health for two gig economy jobs that are absent from this literature.” 

2) More needs to be said about the type/types of sex work participants engaged in. It appears that the authors are making claims/assumptions based on full-service sex work, but this is not made clear. Sex work encompasses more than full-service sex work and the authors need to be clear about the types of sex work research participants did.

We added to text on page 2:

 The sex work sample was limited to people who exchanged a sexual service for money, had one-to-one, direct physical contact with customers, and were paid directly by them. We chose these criteria to focus on the experience of sex workers who provided sexual services to clients in person on at least a part-time basis, in a parallel way that stylists provide in person services to their clients. These interactions occur in a number of contexts ranging from street soliciting to “off-street” work in escorting and erotic massage businesses.

And to text on page 9:

Workers had to have one-to-one, direct physical contact with customers to be considered for the study. Given this parameter, sex work in our study included activities such as escorting, erotic massage, and on street soliciting, but excluded work in which there is no direct physical contact with clients (e.g., phone sex or media productions). We asked participants to name the job title they used to describe their work. For those in sex work, the most common responses were escort, prostitute, sex worker, and sex trade worker.

 3) A major issue here is a claim made that income data for sex workers was among the lower half of income distribution, however, this is not the case for all sex workers – some sex workers have the highest incomes earned and are among the highest half of income distribution, so more clarity is needed.

Our point here was not that all sex work leads to low incomes; indeed, in some cases workers do earn high incomes; instead, our point was about the average or median reported wage in the few studies that have collected income data. We made the following change on page 4 to clarify our point.

 There is much less data on the income of sex workers and the existing data are not easily comparable to the income records for jobs in the formal economy. A study of street-based female sex work in Vancouver, Canada reported median weekly earnings from sex work of $300 CAD or approximately $15,600 CAD a year (Deering et al., 2011); however, the study reported considerable variation in income with a quarter of respondents reporting that they earned less than $100 CAD a week (~$5,200 CAD a year) and a quarter reporting that they earned $560 CAD a week or more (~$29,200 CAD). A heterogeneous sample of sex workers age 19 and over in five Canadian census metropolitan areas found the median total income (earnings and government transfers) for people working in sex work was $39,500 CAD, compared with $34,204 CAD for the general Canadian population aged 15 and over (Authors, 2020), but much lower compared with employment income (wages, salaries and commissions plus net self-employment income) of full-time full-year workers aged 15 and over ( $53,431 CAD; Statistics Canada, 2017).

4) The claim about sex work having low occupational prestige is backed by a very old study from 1980 and is premised on the view of sex work as criminal/a deviant activity. This stands in contradiction to the authors claim that sex work is viewed in the paper as and through a labour lens. Suggest revising this and finding more suitable and current studies and that the authors may wish to undertake a more nuanced and critical analysis.

We could not locate any more recent comparative work on occupational prestige that included sex work and so revised our discussion of SES and occupational prestige on page 4.

Styling (Nakao & Treas, 1994) has low occupational prestige and is at the low end of socio-economic status (SES) scales. Sex work is typically excluded from occupational prestige and SES status research (see Hauser & Warren, 1997).

5) The authors need to be clear about the types of sex work as this is necessary in enabling an understanding of the comparisons made, but also in understanding the importance of laws and legal contexts and so in addition to the above, the authors need to consider the different legislative contexts. Some consideration is given to this, but this is cursory and not enough to fully understand the impacts this may have on workers’ experiences.

We added material on page 6 to address this (also see points 1 and 2 above).

Prohibitions also vary by place, as is evident in the laws from the two places where the data we use in this study were collected. In the U. S., sex work is governed mostly by state law. At the time of our study, and continuing to today, the California Penal Code criminalizes accepting money or other compensation for sex. It penalizes soliciting, as well as engaging in prostitution, as misdemeanors that, after the first conviction, result in mandatory jail time. In Canada, sex work is mostly controlled though the federal criminal code. At the time the data for this study were collected it was legal in Canada to sell and buy sexual services, although it was almost impossible to do so without breaking laws prohibiting solicitation and bawdy houses (Morton et al., 2012). 

6) A major question here is that the US seems to be statistically significant (as indicated in Table 3) – could the different laws be a factor?

The main effect for the US suggests that net of other variables, people in sex work in the US reported better mental health than did those in Canada. We do not know why this is the case but it may be due in part to a larger proportion of the US sample responding to study advertisements posted in health clinics for sex work.   

7) Sex work is illegal in the US, but this was not the case in Canada until 2014 when buying sex was criminalised. The authors need to include a section about the sex work laws in the two different countries, and this needs to include the relevant laws during the periods of data collection. This needs to be taken into account in the analysis as this is likely to have a significant impact on workers’ mental health and job insecurity.

See point 6 above.

8) Here the authors also need to give consideration to the point that not all forms of sex work are full-service (but this can still include direct physical contact) and thus other forms of sex work may be legal in the two research sites where full-service sex work may be criminalised.

See point 2 above.

9) Suggest re-labelling the ‘socially more and less acceptable drug use’ categories to use terms that are less judgmental (e.g., alcohol and marijuana consumption; illicit and non-prescription drug consumption).

We relabeled these simply as substance use 1 and substance use 2, to remove stigmatizing language and also to fit these labels on the tables.

10) The manuscript needs a copy edit, there are a few typos and errors with citations and in the reference list etc. And, detail needs to be provided on the HREC/IRB/IEC review body that approved the project.

We fixed the citations and corrected typos and errors.

We added this text on page 9 about IRB approval:

The research protocol was approved by the IRBs of the authors’ universities (names will be added at end of review process).

Reviewer 2 Report

  1. What proportion of the sex work sample was involved (primarily) in street-based sex work? What proportion was involved (primarily) in escorting? Did these proportions change over the four waves of the study?
  2. 95% confidence intervals should be reported for all Cronbach’s alpha coefficients.
  3. For all measures, it would be helpful if the authors reported the total range of possible scores, and whether higher scores denote more (or less) of the construct of interest.
  4. For the 9-item measure of stigma, was “not applicable” a response option? For example, if a participant had never applied for a bank loan, how would they respond to that item?
  5. The authors should provide separate Cronbach’s alpha coefficients for the skill discretion and decision authority subscales of the 8-item JCQ. Also why didn’t the authors treat these variables as separate predictors rather than using the composite decision latitude variable?
  6. Do the authors have any idea why the 5-item psychological demands subscale of the JCQ had such a poor Cronbach’s alpha coefficient?
  7. For the control variables of sexual orientation and gender identification, the authors should provide all of the response options.
  8. Page 5, lines 201-202: do the 4 correlation coefficients correspond to the four waves of data collection?
  9. What was the response format for the checklist of health enhancing behaviours?
  10. What was the response format for the substance use measure?
  11. Page 3, line 90: One or more words appear to be missing (“It also includes a sizable proportion of independent workers or what some researchers have identified as or entrepreneurs”)
  12. Did the proportions of missing data differ between hairstylists and sex workers?
  13. Table 1: for some of the variables (e.g., live with romantic partner), means are less informative than percentages. (Indeed, for most of the dichotomous “yes/no” variables, percentages would be more easily interpreted by the reader.)
  14. Table 1: for number of children living with the respondent, did the authors first ask if participants had children and, if so, how many of their children lived with them?
  15. Table 1: the authors should also furnish effect sizes for all t-tests and chi-squares.
  16. Table 2: how do the authors interpret the negative beta coefficient for sex work/self-employment (within)?
  17. Table 3: are standardized beta coefficients being reported? (Initially, I thought so but the coefficient for health care use (between) suggests this isn’t the case.)
  18. Page 3, line 130: add missing quotation mark around “dirty”
  19. Page 5, line 194: remove “e” before “in sex work”
  20. Page 5, line 209: use “measured” instead of “measure”
  21. Page 6, lines 228 and 228: should “respectfully” be used instead of “respectively”?
  22. Page 6, line 260: delete “were” before “lived”
  23. Page 7, line 273: remove the bracket after “…activities”
  24. Page 14, line 6: use “self-employed stylists” rather than “self-employment stylists”
  25. Page 15, line 5: use “…economic activity that involves…”
  26. Page 16, line 23: “mangers” should be “managers” (also line 25)
  27. Page 16, line 37: “treating at..” should be “treating it…”
  28. Page 16, line 37: “seeing at…” should be “seeing it…”

Author Response

We thank the reviewers for their helpful suggestions and we have revised our paper accordingly. We provide our responses below, in italics.

Reviewer 2:

1) What proportion of the sex work sample was involved (primarily) in street-based sex work? What proportion was involved (primarily) in escorting? Did these proportions change over the four waves of the study?

We do not have this level of precision in our data. Our pretest showed that it was very unusual for workers to only advertise their services in one location or media such as in a street setting or escort agency website or other advertisement. We used an open-ended question in asking people about their job and we got a diverse array of answers with escort, sex work, and prostitution being the most common. People also reported more than one work arrangement. For example, of the 34 people who described their job as escorting, 16 said they were self-employed, 8 said they were employees, and 10 said they were both. Of the 34 who used sex work or sex trade, 29 said they were self-employed, and 5 said they were both.

2) 95% confidence intervals should be reported for all Cronbach’s alpha coefficients.

We added this information for all scales.

3) For all measures, it would be helpful if the authors reported the total range of possible scores, and whether higher scores denote more (or less) of the construct of interest.

We did this, as well as include Table A1 (page 41-2), which had been inadvertently dropped from our previous submission

4) For the 9-item measure of stigma, was “not applicable” a response option? For example, if a participant had never applied for a bank loan, how would they respond to that item?

Yes. We added this text on page 12to explain this:

Two questions asked respondents how often they had been refused rental housing they could afford or were denied a bank loan (1=Never … 5=Very often; respondents skipped these questions if they indicated they had not applied for rental housing or a loan).

5) The authors should provide separate Cronbach’s alpha coefficients for the skill discretion and decision authority subscales of the 8-item JCQ. Also why didn’t the authors treat these variables as separate predictors rather than using the composite decision latitude variable?

Thank you for your suggestions but we decided not to make these changes. We used the composite measure for three reasons: 1) A Factor analysis shows a one factor solution (Factor1 Eigenvalue= 2.75752; Factor 2 Eigenvalue=0.36463); 2) Our initial analysis showed that the composite scale showed the highest internal validity: alpha for the 8 variable scale is .82; for the 3 variable decision authority subscale it is .66, and for the 5 variable decision latitude sub scale it is .77 (this is consistent with other studies such as the one  we cite by Karasek and colleagues and Niedhammer, I. (2002). International Archives of Occupational and Environmental Health, 75(3), 129-144.); 3) The two components are correlated and share the variance of mental health they account for; thus, neither is significant when entered as two separate terms in the sex work equation.

6) Do the authors have any idea why the 5-item psychological demands subscale of the JCQ had such a poor Cronbach’s alpha coefficient?

No, but the lower alpha is not uncommon. We added this text to page 13:

The third JCQ measure, psychological demands, was a five-item scale (α=.48; 95% CI [.38, .57]; dropping items does not increase alpha) that assessed mental workload and conflicting demands and constraints (Karasek and colleagues (1998) have suggested that the lower internal consistency for this scale in some studies may reflect the greater subjectivity of its items, relative to those in the other scales).

7) For the control variables of sexual orientation and gender identification, the authors should provide all of the response options.

Please see point 3.

8) Page 5, lines 201-202: do the 4 correlation coefficients correspond to the four waves of data collection?

Yes, we added this text on page 10:

We used answers to two questions (r, W1=.66, W2=.65, W3=.73, W4=.66, p<.001) for our overall mental health scale.

9) What was the response format for the checklist of health enhancing behaviours?

See point 3.

10) What was the response format for the substance use measure?

See point 3.

11) Page 3, line 90: One or more words appear to be missing (“It also includes a sizable proportion of independent workers or what some researchers have identified as or entrepreneurs”)

We made this change.

12) Did the proportions of missing data differ between hairstylists and sex workers?

Yes, it was almost all from people in sex work. We added this text to page 15:

The majority of missing cases occurred among respondents who were doing sex work at the first wave of the study.

13) Table 1: for some of the variables (e.g., live with romantic partner), means are less informative than percentages. (Indeed, for most of the dichotomous “yes/no” variables, percentages would be more easily interpreted by the reader.)

We made this change (see page 18).

14) Table 1: for number of children living with the respondent, did the authors first ask if participants had children and, if so, how many of their children lived with them?

We asked respondents a series of questions about their biological children but for this item we focused on the number of children they lived with regardless of their parentage. We added this text to page 14:

We included a control for living with a romantic partner (0=No, 1=Yes) and one for the number of children who lived with the respondent (includes biological, adopted, and children of partners, relatives, or others).

15) Table 1: the authors should also furnish effect sizes for all t-tests and chi-squares.

Change made (see page 17, 18, 19).

16) Table 2: how do the authors interpret the negative beta coefficient for sex work/self-employment (within)?

The 95% confidence interval for this effect is quite large and stretched beyond zero in both directions (-.750, +.183) so we cannot, with any degree of confidence, make conclusions about the direction or size of this association.

17) Table 3: are standardized beta coefficients being reported? (Initially, I thought so but the coefficient for health care use (between) suggests this isn’t the case.)

Unstandardized. We now note this on page 20:

Our multivariable analyses highlighted several patterns (see Table 3). In both sex work and styling, the unstandardized coefficients for job insecurity were significant and negatively related to mental health (b=-.24, se=.10, b= -.35, se=.10, respectively).  

18) Thank you. We made all of the suggested changes below:

Page 3, line 130: add missing quotation mark around “dirty”

Page 5, line 194: remove “e” before “in sex work”

Page 5, line 209: use “measured” instead of “measure”

Page 6, lines 228 and 228: should “respectfully” be used instead of “respectively”?

Page 6, line 260: delete “were” before “lived”

Page 7, line 273: remove the bracket after “…activities”

Page 14, line 6: use “self-employed stylists” rather than “self-employment stylists”

Page 15, line 5: use “…economic activity that involves…”

Page 16, line 23: “mangers” should be “managers” (also line 25)

Page 16, line 37: “treating at..” should be “treating it…”

Page 16, line 37: “seeing at…” should be “seeing it…”

Round 2

Reviewer 1 Report

The author/s have addressed the feedback and the argument is stronger - well done.